# The Effect of Birth Weight on Fattening Performance, Meat Quality, and Muscle Fibre Characteristics in Lambs of the Karayaka Native Breed

**DOI:** 10.3390/ani14050704

**Published:** 2024-02-23

**Authors:** Emre Şirin, Uğur Şen, Yüksel Aksoy, Ümran Çiçek, Zafer Ulutaş, Mehmet Kuran

**Affiliations:** 1Department of Agricultural Biotechnology, Faculty of Agriculture, Kırşehir Ahi Evran University, 40100 Kırşehir, Türkiye; 2Department of Agricultural Biotechnology, Faculty of Agriculture, Ondokuz Mayıs University, 55139 Samsun, Türkiye; mkuran@omu.edu.tr; 3Department of Animal Science, Faculty of Agriculture, Eskişehir Osmangazi University, 26160 Eskişehir, Türkiye; yaksoy@ogu.edu.tr; 4Department of Food Engineering, Faculty of Engineering and Architecture, Tokat Gaziosmanpaşa University, 60100 Tokat, Türkiye; 5Department of Animal Science, Faculty of Agriculture, Ondokuz Mayis University, 55139 Samsun, Türkiye; zulutas@omu.edu.tr

**Keywords:** lamb, birth weight, fattening, muscle fibre, meat quality

## Abstract

**Simple Summary:**

There is little information about the consequences of variation in birth weight due to intrauterine growth on postnatal growth and development of lambs. This study, therefore, investigated the effects of birth weight on postnatal growth, fattening performance, muscle mass development, muscle fibre characteristics, and meat quality in Karayaka lambs. The results of the study show that heavier-born lambs develop better, which is reflected in fattening performance and carcass characteristics. Although there was no difference in meat quality parameters, it was found a significant difference in the fibre area of Type I fibres in the longissimus thoracis et lumborum muscle due to birth weight. The results of the study provide essential insights into the complex relationship between birth weight and various physiological values such as carcass parameters, muscle mass development, and muscle fibre characteristics in post-weaning fattening of Karayaka lambs.

**Abstract:**

This investigation aimed to assess the influence of birth weight on post-weaning fattening performance, meat quality, muscle fibre characteristics, and carcass traits in Karayaka lambs. The study categorized the lambs into three distinct groups based on birth weight: low birth weight (LBW), medium birth weight (MBW), and high birth weight (HBW). Throughout the fattening phase, the lambs were given *ad libitum* access to food and water, culminating in the slaughter at the end of the study. Following slaughter, warm and cold carcasses were weighted, and specific muscles (longissimus thoracis et lumborum [LTL], semitendinosus [ST], and semimembranosus [SM]) were isolated for the evaluation of muscle weights, muscle fibre types (Type I, Type IIA, and Type IIB), and muscle fibre numbers. Carcass characteristics were also determined, including eye muscle (LTL) fat, loin thickness, and meat quality characteristics, such as pH, colour, texture, cooking loss, and water-holding capacity. The statistical analysis revealed highly significant differences among the experimental groups concerning muscle weights and warm and cold carcass weights (*p* < 0.01), with the lambs in the HBW group exhibiting a notably higher carcass yield (in females: 45.65 ± 1.34% and in males: 46.18 ± 0.77%) and LTL, ST, and SM (except for female lambs) muscle weights than the lambs in LBW group (*p* < 0.01). However, apart from the texture of LTL and ST muscles, no significant differences in meat quality parameters were observed among the treatment groups (*p* > 0.05). Notably, the birth weight of lambs did not impart a discernible effect on the total number and metabolic activity of muscle fibres in LTL, ST, and SM muscles. Nonetheless, a noteworthy distinction in the fibre area of Type I fibres in the LTL muscle of male lambs (LBW: 30.4 ± 8.9, MBW: 29.1 ± 7.3 and HBW; 77.3 ± 15.4) and in the ST muscle of female lambs (LBW: 44.1 ± 8.1, MBW: 38.8 ± 7.7 and HBW: 36.9 ± 7.1) were evident among the birth weight groups (*p* < 0.05). The study also found that the mean fat thickness values of eye muscles in Karayaka lambs, as obtained by ultrasonic tests, were below the typical range for sheep. In synthesis, the outcomes of this study underscore the considerable impact of birth weight on slaughtered and carcass weights, emphasizing the positive association between higher birth weights and enhanced carcass yield. Remarkably, despite these pronounced effects on carcass traits, the birth weight did not demonstrate a statistically significant influence on meat quality or overall muscle fibre characteristics, except for the area of Type I fibres in the LTL muscle. This nuanced understanding contributes valuable insights into the intricate relationship between birth weight and various physiological and carcass parameters in Karayaka lambs undergoing post-weaning fattening.

## 1. Introduction

Lamb production is one of the most critical parts of sheep farming. Developing countries raise different domestic sheep breeds tailored to the region. There are over 42 million sheep in Türkiye [1], and the survival of the sheep business depends heavily on the advancement of the local breeds. Among these native breeds, the Karayaka sheep breed is intensively bred in the Black Sea region. They are a carpet-wool breed also kept for meat production. Karayaka is well adapted to harsh environmental conditions such as poor climates and pasture [2].

Birth weight is the primary factor influencing the newborn’s viability and long-term health [3,4], and knowing the controllable factors affecting live weight is essential to the agricultural economy. All mammalian species have an uncomplicated birth and a birth weight that maximizes the newborn’s viability. When birth weight deviates from this optimum, there is only a range in which the newborn will survive until reproductive age. Certain genetic factors are responsible for variations in birth weight. While low birth weight causes an increase in neonatal mortality, high birth weight can lead to an increase in difficult birth or maternal mortality [5].

Maternal nutrition level during pregnancy significantly affects the weight gain of the placenta and fetus, altering the birth weight of the offspring [6], while its level in the middle and late periods of pregnancy affects the offspring’s postnatal skeletal muscle fibre number and composition [7,8]. Transitioning from the fetal period to postnatal life requires the maturation of biological systems necessary for survival and growth [9]. Appropriate environmental and nutritional conditions may allow healthy but low-birth-weight lambs to survive and grow rapidly [10]. However, the period before the onset of rapid growth may be longer in low-birth-weight lambs, indicating that their metabolic systems require a more extended period to adapt to postnatal life [11]. Prolonging the adaptation period necessary for survival after birth may affect postnatal productivity, skeletal muscle growth, the formation of body structures such as fat and connective tissue, and meat yield and even quality [8,11,12,13].

According to Barker’s “Fetal Origin of Adult Diseases” hypothesis, the size of the placenta in the fetal period affects the amount of nutrients transferred to the fetus. Adequate feeding in the early period of pregnancy increases the size of the placenta; the increase in placental size means more nutrients are available or transferred to the fetus, and thus the birth weight increases [14]. The first half of the gestation period, in which muscle fibres are programmed in the fetus, coincides with the decrease in grass growth in pasture areas in the northern hemisphere. Therefore, pasture grazing may not fully meet maternal nutritional needs during gestation in such cases. Maternal malnutrition may adversely affect the development of the fetus and cause a wide variation in birth weight [15]. Previous studies showed that maternal nutrition level in the critical periods of pregnancy (for example, between the 30th and 80th days) in sheep and cattle affects birth weight and postpartum muscle development and fattening performance [7,16,17].

The prenatal period is critical for muscle fibre development since it involves an increase in the number of muscle fibres. There is, however, no apparent increase in the number of muscle fibres in the postpartum period [18]. The feeding strategies applied in the early stages of this critical prenatal period when the muscle fibres are formed may change the muscles’ cellular activity and subsequently affect the meat quality depending on the birth weight, postpartum muscle development, and slaughter weight. In sheep, hyperplasia of fetal muscle fibres begins at ~30 or 32 days and is completed at ~85–90 days gestation [19]. Prenatal development of muscle fibres may affect the meat quality obtained when animals reach adult body weight. The characteristics of the skeletal muscle tissue vary according to breed [20], sex [21], hormone level [22], growth term [23], nutrition [8], and muscle location [24]. The intrinsic properties of the muscle vary according to the muscle fiber composition of the skeletal muscle tissue. This change determines the meat’s colour, odour, flavour, juiciness, tenderness, and texture [25]. The metabolic and contractile properties of skeletal muscle tissue vary depending on the muscle fibre composition [20,21]. The determining relationship between muscle fibres and meat quality characteristics will contribute to the programming of better-quality meat production. The total number, diameter, type, and composition of muscle fibers forming the muscle are essential in determining meat quality after slaughter [20]. Only a few studies have simultaneously assessed the meat quality and muscle fibre characteristics in Karayaka lambs. This study aimed to ascertain how differing birth weights affected the ability of the Karayaka lambs to fatten, the features of the carcass, some meat quality parameters, and the muscle fibre characteristics.

## 2. Materials and Methods

### 2.1. Animals

The experiment was conducted at the Agricultural Research Unit of Gaziosmanpasa University in Tokat, Türkiye, during the breeding season (September to March) (40031′ N, 36053′ E, 650 m above sea level). Forty-three newborn lambs, born to Karayaka ewes at least in the second parturition and ranging from 2 to 3 years of age, were dried and weighed before suckling. Lambs were then divided into three groups: low birth weight (LBW; <3.59 kg), medium birth weight (MBW; 3.59–4.89 kg), and high birth weight (HBW; >4.89 kg). Birth weight groups were determined as lambs with one standard deviation difference of the average considering the average weight of all lambs born in the same flock and period. Table 1 presents the distribution numbers of lambs by birth weight and sex.

### 2.2. Growth and Fattening of Lambs

All lambs were kept in the pen with dams for two weeks after lambing and allowed to suckle their dams freely. The ewes were fed an average of 200 g concentrate (15% crude protein, 2800 kcal/kg dry matter metabolizable energy) and alfalfa of 1 kg per day for two weeks. Ewes were allowed to pasture for daytime grazing two weeks after birth and suckle their lambs overnight in sheepfold until weaning. In addition to pasture grazing, the ewes were fed an average of 150 g of concentrate per day during lactation. When all lambs were two months old, they were treated with protection against internal and external parasites and introduced to a creep-feed concentrated diet (15% crude protein, 2500 kcal/kg dry matter metabolizable energy) fed *ad libitum* with good quality alfalfa as a supplement to ewe’s milk. At three months of age, all lambs were weaned, and feed and water were withheld overnight to assess fasting body weight the next day at weaning. After weaning, all lambs were subjected to a two-month fattening period and slaughtered at five months of age. The lambs were fed a mixed diet of *ad libitum* alfalfa hay and concentrated feed during the fattening period. Water and mineral stones were freely available during the fattening period. The nutrient contents of experimental feeds were provided by the manufacturer (Güven Yem A. Ş., Çorum, Türkiye) and shown in Table 2.

### 2.3. Measurements and Muscle Sample Collection

At the end of the fattening period, feed and water were withdrawn overnight to determine fasting body weight the following day. All lambs were transported to a local slaughterhouse after being weighed, and standard commercial slaughter procedure was carried out. Before slaughter, the loin thickness of eye muscle of longissimus thoracis et lumborum muscle (LTL) was measured in all lambs by an ultrasonic linear prop (Falco Vet. Linear prop 8.0 MHz; Pie et al., The Netherlands) as described by Ulutas et al. [2]. The pelt, head, feet, internal organs (spleen, lungs, liver, and kidney), empty reticulorumen, and empty intestine were weighed after slaughter. Warm carcass weights were also measured after removing all internal organs. Immediately after slaughter, LTL, semitendinosus (ST), and semimembranosus (SM) skeletal muscles were isolated and weighed from the right side of the carcasses. Two samples measuring of 5 × 2 × 2 cm (approximately 50 g) were taken from the mid-sections of LTL, ST, and SM muscles. Fat and connective tissue were trimmed from muscle samples, immediately covered with aluminium foil, frozen in liquid nitrogen, and stored at −80 °C until histochemical analysis of muscle fibres. In addition to internal fat, pelvic and kidney fat were isolated and weighed. All carcasses with isolated muscles were chilled for 24 h at 4 °C. To determine meat quality traits, approximately 150–200 g muscle samples were collected from the central parts of the mid-section of the whole LTL and ST muscles from the left side of the carcasses after 24 h chilled storage. The fat and connective tissue of LTL and ST muscle samples were trimmed before meat quality analysis and storage at 4 °C.

### 2.4. Meat Quality Analyses

Tenderness, pH, and colour characteristics of LTL and ST muscle samples were specified as described by Şen et al. [28]. The pH value was determined using the meat pH meter with a puncture electrode (Testo 205, Lenzkirch, Germany) at 1 and 24 h postmortem. Meat colour as L* (lightness), a* (redness) and b* (yellowness) value was determined using a chromometer (Konica Minolta CR-300, Minolta Co., Ltd., Osaka, Japan) at 1 and 24 h postmortem. The water-holding capacity of LTL and ST muscle samples (approximately 25 g) was specified by the filter paper press method [29] with some modifications. About 3.0 g of intact muscle sample was weighed and placed on a previously desiccated and weighed filter paper (Whatman No. 1, 11 cm in diameter) with two thin plastic films. After weighing the meat sample, the filter paper and plastic film with the meat sample were placed between wood plates. Loads of 2.5 kg were applied for 5 min using weights. After accurately removing the compressed meat sample, it was rapidly weighed to determine the percentage water-holding capacity of the meat sample. Cooking loss was analyzed as described by Mitchaothai et al. [30] in LTL and ST muscle samples. Approximately 50.0 g muscle samples were put in a plastic bag and then cooked for 40 min in a water bath at 70 °C constant temperatures. The samples were then exposed to running water for approximately 15 min to cool to room temperature. The samples were weighed before and after cooking to determine the percentage of cooking loss. 

### 2.5. Histochemical Determination of Muscle Fibre Type Composition

Contractile types (Type I, IIA, and IIB) of muscle fibres in LTL, ST, and SM muscles were analyzed using myosin ATPase staining at pH 4.2, as described by Broke and Keiser [31] and Şen et al. [8]. Metabolic types (oxidative and glycolytic) of muscle fibres in LTL, ST, and SM muscles were determined using the succinate dehydrogenase (SDH) activity staining described by Nachlas et al. [32]. Six transverse serial muscle sections (10 mm thick) from muscle samples were obtained using a cryostat (Microtome E, Thermo Electron Corporation, Basingstoke, UK) at −20 °C. Sections were allowed to dry; three of the muscle sections were used for ATPase staining, while the remaining three were used for SDH activity staining. Muscle fibres were counted using a microscope (Nikon Eclipse E600, Nikon Corporation, Tokyo, Japan) linked to an image analysis software (Laica Q Win V3.4 Processing-Analysis Software). Four areas were selected randomly from the sections to determine fibre type composition and myofibre diameter. Myofibre diameter was measured from ~25 fibres of each fibre type from each area coun ted. Figure 1 and Figure 2 present pictures of stained muscle fibres from LTL, ST, and SM muscles for myosin ATPase (pH 4.2) and SDH activity staining, respectively.

### 2.6. Statistical Analyses

The effect of birth weight on the postnatal growth rate, carcass composition, muscle fibre characteristics, meat quality of lambs, and other traits was analyzed as a complete randomized design using the general linear model of the Statistical Analysis System [33]. One-way ANOVA was used to compare the gender within each birth weight group and to compare initial and final pH and color characteristics. Tukey’s multiple comparison tests compared the differences in the mean values, and results were computed as mean ± standard error of the mean (SEM). Relationships between birth weight and other traits were determined by Pearson’s correlation analysis. Statistical significance was considered at *p* < 0.05.

## 3. Results

The results of Pearson correlation analysis between birth weight and other traits were not significant (*p* > 0.05). Table 3 shows the results of weaning weight, slaughter weight, daily weight gain, eye muscle values, and carcass characteristics of Karayaka lambs with different birth weights. The study found no significant differences among birth weight groups regarding weaning weight in female lambs, but HBW male lambs had higher (*p* < 0.05) weaning weight than LBW and MBW male lambs. Slaughter weight and total weight gain in HBW female lambs were higher (*p* < 0.05) than in LBW female lambs. Similarly, HBW male lambs had higher (*p* < 0.05) slaughter weights than LBW and MBW male lambs. Although the total weight gain of MBW male lambs was higher (*p* < 0.05) than LBW male lambs, the birth weights of male and female lambs were similar in the LBW, MBW, and HBW groups. Significant differences in the weaning weight (except for the weaning weight of LBW lambs), slaughter weight, and total weight gain were detected between male and female lambs (*p* < 0.05) in birth weight groups. However, there were no significant correlations among birth weight, fattening performance, and carcass characteristics in male and female lambs. There were no significant differences among birth weight groups in fat thickness above the eye muscle, but female lambs had higher (*p* < 0.05) fat thickness than male lambs in each birth weight group. Although loin thickness values of the eye muscle in female lambs were similar in birth weight groups, HBW male lambs had higher (*p* < 0.05) loin thickness than LBW male lambs. Moreover, male lambs in MBW and HBW groups had thicker (*p* < 0.05) eye muscles than female lambs in the same birth weight groups. In addition, the difference among the birth weight groups in terms of LTL, ST, SM (except for the SM muscle of female lambs) muscle weights, warm, cold carcass weights, and yields were significant (*p* < 0.01). Additionally, MBW and HBW male lambs had more significant LTL and SM muscle weights (*p* < 0.05) than female lambs in the same birth weight groups (except for the SM muscle of MBW lambs). Warm and cold carcass weights of male lambs were significantly higher than female lambs (*p* < 0.05) in each birth weight group (except for the cold carcass of LBW lambs).

Table 4 shows several traits of the meat quality in the LTL and ST muscles of female and male Karayaka lambs at various birth weights. The study found no significant differences among the birth weight groups for meat quality attributes, including water-holding capacity, and cooking loss in LTL and ST muscles (*p* > 0.05). There were no significant differences among birth weight groups in texture values of ST muscle in both sexes, but HBW female lambs had higher (*p* < 0.05) texture values than LBW female lambs in LTL muscle. Although there were no significant differences among birth weight groups in pH values of LTL and ST muscles in both sexes, the pH values measured at 1st hour in both muscles of female and male lambs in the birth weight groups were higher than those measured at the 24th hour (*p* < 0.05). The water-holding capacity values of LTL and ST muscles in female lambs were higher (*p* > 0.05) than in male lambs in each birth weight group. Similarly, cooking loss values of LTL muscle in female lambs were higher (*p* > 0.05) than in male lambs in each birth weight group. Table 5 also presents colour measurement (L*, a*, and b*) values in LTL and ST muscles at the 1st hour and 24th hour after slaughtering Karayaka female and male lambs at different birth weights. There was no difference in the L, a, and b colour values in the LTL and ST muscles among the birth weight groups after the 1st and 24th hours (*p* > 0.05). The data presented in Table 5 also indicate that there was no significant difference in the mean L*, a*, and b* values of the LTL and ST muscles from male and female lambs at the 1st and 24th hours postmortem. Within the same birth weight groups, however, the variations in the colour averages (L*, a*, and b* values) were significant at the 1st and 24th hours (*p* < 0.05).

Table 6 shows the area of muscle fibres in LTL, ST, and SM muscles of Karayaka lambs at different birth weights. In the LTL muscle, there was a significant difference (*p* < 0.05) in the area of Type I muscle fibres among the birth weight groups in male lambs; lambs with high birth weight had the largest Type I muscle fibre area (*p* < 0.05). Additionally, male lambs in the HBW group had larger (*p* < 0.05) Type I muscle fibre areas than female lambs in the same birth weight group. Similarly, female lambs with high birth weight had the largest Type I muscle fibre area (*p* < 0.05) in ST muscle. There was, however, no difference among the birth weight groups and sexes in terms of other muscle fibre (Type IIA and IIB) areas. The data for different muscle fibre numbers in LTL, ST, and SM muscles are presented in Table 7. There was no statistical difference in Type I, IIA, and IIB muscle fibre numbers among the birth weight groups. Moreover, the metabolic activity ratios of muscle fibre in LTL, ST, and SM muscles were similar. However, male lambs had higher (*p* < 0.05) Type IIA muscle fibre numbers in LTL muscle than female lambs in each birth weight group. Additionally, LBW male lambs had higher (*p* < 0.05) Type IIB and total muscle fibre numbers than female lambs in LTL muscle. However, no significant correlations existed between birth weight and muscle fibre characteristics in male and female lambs.

## 4. Discussion

Birth weight is one of the most important factors affecting the growth of the newborn. Sušić et al. [34] reported that there is a relationship between birth weight, subsequent live weight development, and fattening performance. The present study underscores the observation that lambs within the high birth weight category attained elevated slaughter weights. It has been determined that birth weight significantly affects slaughter weight in Karayaka lambs. This aligns with the findings by Şen et al. [28], who reported a slaughter weight of 35.6 kg in a corresponding study. In contrast, the slaughter weights recorded in both MBW and HBW groups in the current investigation surpassed those reported by Şen et al. [28]. According to Boggs and Merkel [35], the typical range of the area of the eye muscle and the depth of fat (fat thickness) above the 12th and 13th rib in lambs following fattening should be between 3.81 and 9.14 cm^2^ and 0.508 and 1.27 cm, respectively. The current study found that the mean fat thickness values of Karayaka lambs, as determined by ultrasonic tests, were below the previously stated range [35]. In this investigation, higher weights of both warm and cold carcasses were observed in the HBW group. The male Karayaka lambs in the HBW group exhibited higher weights both warm and cold carcasses, surpassing Şen et al.’s [28] documented weights for the same breed. In contrast, studies by Greenwood et al. [36,37] suggest that birth weight may alter the postnatal growth of individual organs and some skeletal muscle mass without influencing their combined weight at equivalent body weights. Although there was no relationship between birth weight and carcass characteristics in the current study, our observation of carcass weight increase as birth weight increases emphasizes that the effect of birth weight on post-fattening carcass weights should be evaluated. The greater fattening weights of lambs with high birth weights can be considered the cause of birth weight variance. The greater fattening weights of lambs with high birth weights can be considered the cause of birth weight variance in the present study. 

The ability of fresh meat to retain water following rigour mortis and its texture are two crucial meat quality attributes significantly impacted by its pH levels. According to Aksoy et al. [38], meat quality is, therefore, greatly influenced by the pH value. Sañudo et al. [39] found that there might be notable variations in pH levels across different sheep breeds. For the same race, however, this distinction could not be acknowledged. Texture measures at the 24th and 1st hours marks exhibited no conspicuous variations. The final pH levels between 5.8 and 6.0 reduce the sensory tenderness score [40]. The postmortem pH values of LTL and ST muscles in lambs of varying birth weights were under 5.8 and greater than 6.0, respectively. Essentially, the physical qualities of meat, such as water-holding capacity and cooking loss, influence the yield and quality of meat products. Water-holding capacity and cooking loss values are also connected to postmortem biochemical features such as proteolysis, muscle protein shrinkage (actin and myosin), and cell wall breakdown. These biological processes influence intercellular water release. The water-holding capability of meat increases qualities such as tenderness and juiciness [38]. The water-holding capacity of the ST and LTL muscles was lower in this study than in previous studies by Aksoy et al. [38] and Şen et al. [28].

Furthermore, the colour of meat serves as a crucial indicator of product freshness, with consumers preferring light red or pink lamb meat [41]. This study showed no variations in L*, a*, and b* values among the birth weight groups in ST and LTL muscles at the 1st and 24th hours. On the other hand, the average L*, a*, and b* values between the 1st and 24th hours were significantly different in each birthweight group. According to Ugurlu et al. [42], the values of L*, a*, and b*, as determined at the postmortem 24 h point in sheep breeds in Türkiye, ranged from 37.91 to 42.72, 16.08 to 21.26, and 5.60 to 8.45, respectively. The 24th hour values of L*, a*, and b* were all within the acceptable ranges.

In line with previous research, a recent study by Şen et al. [8] outlined that the maternal nutrition level during the mid-gestation period (from 30 to 80 days), during which skeletal muscle fibres partially begin and complete the growth and development, has a significant impact on the offspring’s growth performance, carcass composition, fibre types, muscle fibre muscle area in mm^2^, and growth (diameter) of skeletal muscle fibre [7,16,17]. Despite the absence of a discernible increase in the number of muscle fibres during the postpartum period, the prenatal period is crucial for developing muscle fibres since this is the period when they proliferate and differentiate [18]. The quantity and type of muscle fibres in a lamb remain unaffected by environmental factors, including feeding, movement, and maintenance throughout the postnatal stage. However, these factors, particularly nutrition, impact muscle fibre diameter. Fahey et al. [7] and Dwyer et al. [43] suggested that a significant increase in muscle fibre diameter may result from a reduced total number of muscle fibers. Nevertheless, in the present study, muscle fibre compositions and numbers remained unaffected by birth weight in both muscles and sexes. These results were in contrast to previous studies in which the mean fibre number in ST muscle of high birth weight lambs was higher than that of low-birth-weight female lambs [44]. The finding that muscle fibre number was unaffected by birth weight, despite the LBW lambs having nearly 65% of the weight of HBW lambs at birth, aligns with the results of Greenwood et al. [36]. In the present study, ewes were exposed to similar environmental conditions during the critical period for prenatal muscle fibres. However, differences were detected in the Type I muscle fibres in the LTL muscle of male lambs and the ST muscle of females due to the disparity in the birth weight of the lambs.

The differentiation between muscle fibres based on their contraction speed is achieved using the histochemical test for myofibrillar ATPase activity. Measures of ATPase activity can be interpreted in terms of contraction speed since myosin ATPase activity and muscle contraction velocity are positively associated [18]. Similarly, the histochemical test for SDH activity staining distinguishes oxidative and nonoxidative (or, more accurately, “less” oxidative) fibres. High-oxidative fibres use oxidative phosphorylation in the mitochondria to produce ATP, and muscle cells with a higher number of mitochondria will have a greater capacity for oxidation. The SDH enzyme, situated in the inner membrane of the mitochondrion and attached to the cristae, oxidizes succinate to fumarate in the citric acid cycle. Succinate is oxidized during this process, resulting in the production of NADH in its reduced form. The staining intensity increases with the amount of SDH (and hence, mitochondrial) activity a fibre contains, as does the ATPase assay. Compared to nonoxidative fibres, which only feature sporadic purple speckles, oxidative fibres exhibit a more densely speckled appearance. In this investigation, the numbers of Type I, IIA, IIB, and total muscle fibres in LTL, ST, and SM muscles were similar across all the weight groups for both sexes. The metabolic activity ratios for each muscle group were also the same for all groups, as expected for individuals of the same breed.

## 5. Conclusions

In conclusion, this study underscores the significant impact of birth weight on slaughter weight in lambs, and this implies that higher birth weight may contribute to increased carcass yield, a crucial consideration in the sheep industry. Interestingly, while birth weight emerged as a pivotal factor influencing carcass weights after fattening, it did not exhibit a discernible effect on meat quality characteristics. These results suggest that, within the parameters studied, birth weight may not directly determine meat quality in lambs. Notably, the study identified a specific area of impact related to muscle fibre types, as an increase in Type I muscle fibre area was observed in lambs with higher birth weights. This finding may contribute to understanding the complex interplay between birth weight and muscle fibre development. This study informs future strategies for optimizing carcass yield in the context of birth weight variations.

## Figures and Tables

**Figure 1 animals-14-00704-f001:**
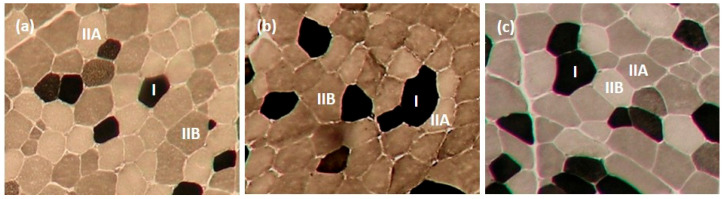
Contractile types of muscle fibre identified by Myosin ATPase staining in longissimus thoracis et lumborum (**a**), semitendinosus (**b**), and semimembranosus (**c**) muscles (10×). The darkest muscle fibre is Type I, the intermediate muscle fibre is Type IIB, and the lightest muscle fibre is Type IIA.

**Figure 2 animals-14-00704-f002:**
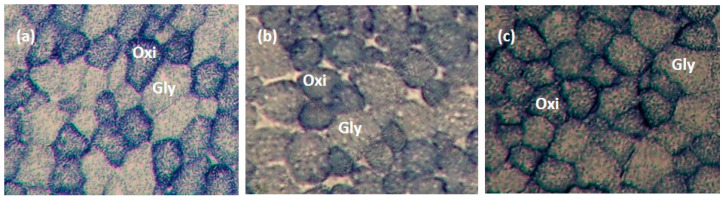
Metabolic types of muscle fibre were identified by succinate dehydrogenase (SDH) activity staining in longissimus thoracis et lumborum (**a**), semitendinosus (**b**), and semimembranosus (**c**) muscles (10×). The darkest blue muscle fibre is oxidative (oxi; Type I), and the lightest blue muscle fibre is glycolytic (gly; Type IIA and Type IIB).

**Table 1 animals-14-00704-t001:** The distribution numbers of lambs by sex.

Sex	Birth Weight (kg)		
	<3.59	3.59–4.89	>4.89
Male	7	7	7
Female	8	7	7

**Table 2 animals-14-00704-t002:** Nutrient contents of concentrate and alfalfa hay (% on dry matter basis).

Nutrient	Concentrate	Alfalfa Hay
Dry matter	90.84	94.02
Crude protein	18.12	15.01
Crude fiber	6.40	24.70
Crude fat	3.69	0.74
Crude ash	7.32	10.30
Metabolizable energy (kcal/kg dry matter) *	2736.00	1878.00

* Metabolizable energy (ME) values of concentrate feed and alfalfa hay were calculated by following prediction equations described by Alderman [26] (ME (kcal/kg dry matter) = ((11.78 + (0.00654 × crude protein %) + (0.000665 × crude fat %)^2^ − (0.00414 × crude fat %) × crude fiber % − (0.0118 × crude ash %))/4.184) × 1000), and Menke and Steingass [27] (ME (kcal/kg dry matter) = 1.68 + (0.1418 × gasses production) + (0.073 × crude protein %) + (0.217 × crude fat %) − (0.025 × crude ash %) × 1000), respectively.

**Table 3 animals-14-00704-t003:** Fattening performance and carcass characteristics of Karayaka lambs.

Traits	Sex	LBW (F; *n* = 8 and M; *n* = 7)	MBW (F; *n* = 7 and M; *n* = 7)	HBW (F; *n* = 7 and M; *n* = 7)	Sig.
Birth weight (kg)	F	2.95 ± 0.08 ^c^	3.78 ± 0.11 ^b^	4.78 ± 0.15 ^a^	0.008
	M	3.13 ± 0.07 ^c^	3.93 ± 0.09 ^b^	5.01 ± 0.16 ^a^	0.012
Fattening performance					
Weaning weight (kg)	F	18.05 ± 0.87	18.51 ± 0.92 ^B^	19.94 ± 0.91 ^B^	0.653
	M	19.03 ± 0.85 ^c^	22.29 ± 0.93 ^bA^	27.68 ± 1.03 ^aA^	0.032
Slaughter weight (kg)	F	26.66 ± 1.51 ^bB^	28.91 ± 0.93 ^abB^	31.64 ± 1.18 ^aB^	0.038
	M	31.91 ± 0.71 ^cA^	37.37 ± 1.12 ^bA^	42.12 ± 1.31 ^aA^	0.027
Total weight gain (kg)	F	8.65 ± 0.49 ^bB^	10.43 ± 1.01 ^abB^	11.72 ± 0.75 ^aB^	0.041
	M	12.91 ± 0.93 ^bA^	15.10 ± 0.69 ^aA^	14.44 ± 1.20 ^abA^	0.043
Eye muscle values					
Fat thickness (cm)	F	0.11 ± 0.02 ^A^	0.11 ± 0.03 ^A^	0.14 ± 0.03 ^A^	0.329
	M	0.05 ± 0.01 ^B^	0.06 ± 0.02 ^B^	0.05 ± 0.02 ^B^	0.427
Loin thickness (cm)	F	0.33 ± 0.08	0.34 ± 0.13 ^B^	0.36 ± 0.12 ^B^	0.365
	M	0.35 ± 0.1 ^b^	0.41 ± 0.08 ^abA^	0.52 ± 0.09 ^aA^	0.039
Carcass characteristics					
LTL muscle weight (g)	F	144.19 ± 8.13 ^b^	164.50 ± 8.85 ^abB^	174.14 ± 6.72 ^aB^	0.046
	M	158.73 ± 6.80 ^c^	187.20 ± 11.20 ^bA^	207.14 ± 5.55 ^aA^	0.011
ST muscle weight (g)	F	114.70 ± 8.57 ^b^	128.7 ± 11.4 ^b^	152.00 ± 5.25 ^a^	0.037
	M	118.60 ± 11.00 ^b^	132.60 ± 13.90 ^b^	166.36 ± 6.35 ^a^	0.039
SM muscle weight (g)	F	38.50 ± 2.92	42.72 ± 1.46	43.00 ± 1.37 ^A^	0.457
	M	42.70 ± 2.70 ^b^	46.60 ± 2.90 ^b^	55.9 ± 2.20 ^aB^	0.041
Warm carcass weight (kg)	F	11.52 ± 0.83 ^bB^	13.72 ± 0.50 ^abB^	14.25 ± 0.50 ^aB^	0.038
	M	13.57 ± 0.49 ^cA^	16.74 ± 0.83 ^bA^	19.74 ± 0.45 ^aA^	0.013
Cold carcass weight (kg)	F	11.36 ± 0.87 ^b^	13.01 ± 0.54 ^aB^	13.96 ± 0.50 ^aB^	0.036
	M	12.94 ± 0.46 ^c^	15.67 ± 0.74 ^bA^	18.58 ± 0.37 ^aA^	0.014
Carcass yield (%)	F	42.55 ± 0.96 ^b^	44.60 ± 0.58 ^ab^	45.65 ± 1.34 ^a^	0.036
	M	42.25 ± 0.96 ^c^	44.95 ± 0.67 ^b^	46.18 ± 0.77 ^a^	0.009

^a,b,c^ Means with different letters in the same rows between birth weight groups are significantly different at *p* < 0.05. ^A,B^ Means with different letters in the same column between sexes for each trait are significantly different at *p* < 0.05. Sig. = *p*-value of significant differences, LBW = low birth weight, MBW = medium birth weight, HBW = high birth weight, F = female, M = male.

**Table 4 animals-14-00704-t004:** Meat quality characteristics in LTL and ST muscles.

Traits	Muscles	Time (h) *	Sex	LBW (F; *n* = 8 and M; *n* = 7)	MBW (F; *n* = 7 and M; *n* = 7)	HBW (F; *n* = 7 and M; *n* = 7)	Sig.
pH	LTL	1	F	6.59 ± 0.08	6.50 ± 0.06	6.41 ± 0.09	0.642
			M	6.42 ± 0.07	6.51 ± 0.04	6.43 ± 0.08	0.585
		24	F	5.53 ± 0.04	5.52 ± 0.03	5.55 ± 0.02	0.662
			M	5.78 ± 0.05	5.72 ± 0.03	5.71 ± 0.03	0.654
	ST	1	F	6.59 ± 0.10	6.57 ± 0.07	6.58 ± 0.06	0.854
			M	6.46 ± 0.09	6.50 ± 0.06	6.54 ± 0.12	0.754
		24	F	5.54 ± 0.02	5.53 ± 0.02	5.54 ± 0.01	0.812
			M	5.79 ± 0.05	5.79 ± 0.03	5.74 ± 0.06	0.954
WHC (%)	LTL		F	6.79 ± 0.78 ^A^	7.56 ± 0.59 ^A^	7.24 ± 0.82 ^A^	0.442
			M	4.79 ± 0.45 ^B^	3.38 ± 0.37 ^B^	4.28 ± 0.81 ^B^	0.503
	ST		F	8.27 ± 0.93 ^A^	8.73 ± 0.72 ^A^	8.00 ± 0.74 ^A^	0.804
			M	3.60 ± 0.47 ^B^	3.10 ± 0.30 ^B^	3.64 ± 0.34 ^B^	0.754
CL (%)	LTL		F	25.44 ± 1.16 ^A^	21.84 ± 1.25 ^A^	21.63 ± 2.06 ^A^	0.798
			M	12.79 ± 0.66 ^B^	14.31 ± 1.61 ^B^	14.79 ± 2.16 ^B^	0.212
	ST		F	24.36 ± 1.01	24.65 ± 1.54	24.35 ± 1.87	0.697
			M	22.29 ± 1.54	21.77 ± 2.13	22.41 ± 1.76	0.831
Texture (kg/cm^2^)	LTL		F	2.44 ± 0.08 ^b^	2.74 ± 0.07 ^ab^	3.09 ± 0.25 ^a^	0.039
			M	3.67 ± 0.19	3.58 ± 0.21	3.69 ± 0.22	0.798
	ST		F	3.73 ± 0.19	3.74 ± 0.19	4.06 ± 0.17	0.691
			M	3.75 ± 0.18	4.20 ± 0.24	3.81 ± 0.18	0.432

^a,b^ Means with different letters in the same rows between birth weight groups are significantly different at *p* < 0.05. ^A,B^ Means with different letters in the same column between sexes for each trait are significantly different at *p* < 0.05. * Means of initial and final pH values for each sex in birth weight groups are significantly different at *p* < 0.05. WHC = water-holding capacity, CL = cooking loss Sig. = *p*-value of significant differences, LBW = low birth weight, MBW = medium birth weight, HBW = high birth weight, F = female, M = male, LTL = longissimus thoracis et lumborum, ST = semitendinosus.

**Table 5 animals-14-00704-t005:** Colour values in LTL and ST muscles ^†^.

Colour	Muscles	Time (h)	LBW (*n* = 15)	MBW (*n* = 15)	HBW (*n* = 15)
L*	LTL	1	34.95 ± 0.59 ^b^	33.76 ± 0.40 ^b^	34.81 ± 0.42 ^b^
		24	41.20 ± 0.29 ^a^	40.82 ± 0.56 ^a^	42.53 ± 0.73 ^a^
	Sig.		0.042	0.041	0.039
	ST	1	32.91 ± 0.58 ^b^	32.19 ± 0.28 ^b^	32.82 ± 0.41 ^b^
		24	39.64 ± 0.72 ^a^	39.61 ± 0.48 ^a^	40.10 ± 0.47 ^a^
	Sig.		0.041	0.041	0.038
a*	LTL	1	17.52 ± 0.32 ^b^	17.47 ± 0.27 ^b^	17.62 ± 0.37 ^b^
		24	19.72 ± 0.28 ^a^	20.37 ± 0.37 ^a^	20.27 ± 0.37 ^a^
	Sig.		0.048	0.047	0.046
	ST	1	18.74 ± 0.26 ^b^	19.25 ± 0.24 ^b^	19.47 ± 0.25 ^b^
		24	19.81 ± 0.33 ^a^	20.02 ± 0.34 ^a^	20.73 ± 0.31 ^a^
	Sig.		0.049	0.047	0.046
b*	LTL	1	4.14 ± 0.23 ^b^	4.25 ± 0.25 ^b^	4.01 ± 0.20 ^b^
		24	6.99 ± 0.23 ^a^	6.50 ± 0.15 ^a^	6.65 ± 0.18 ^a^
	Sig.		0.036	0.037	0.032
	ST	1	4.89 ± 0.21 ^b^	4.89 ± 0.19 ^b^	4.74 ± 0.18 ^b^
		24	6.80 ± 0.16 ^a^	6.45 ± 0.22 ^a^	6.67 ± 0.16 ^a^
	Sig.		0.041	0.043	0.039

^†^ The effect of sex on colour values were not significant, and thus, the mean values of sexes for both muscles were pooled. ^a,b^ Means of initial and final colour values for each sex in birth weight groups are significantly different at *p* < 0.05. Sig. = *p*-value of significant differences, LBW = low birth weight, MBW = medium birth weight, HBW = high birth weight, F = female, M = male, LTL = longissimus thoracis et lumborum, ST = semitendinosus.

**Table 6 animals-14-00704-t006:** Muscle fibre area (μm^2^) in LTL, ST, and SM muscles.

Muscles	Fibre Types	Sex	LBW (F; *n* = 8 and M; *n* = 7)	MBW (F; *n* = 7 and M; *n* = 7)	HBW (F; *n* = 7 and M; *n* = 7)	Sig.
LTL	I	F	40.2 ± 15.3	44.1 ± 15.3	47.0 ± 9.2 ^B^	0.295
		M	30.4 ± 8.9 ^b^	29.1 ± 7.3 ^b^	77.3 ± 15.4 ^aA^	0.021
	IIA	F	35.3 ± 8.0	34.3 ± 7.5	42.0 ± 13.2	0.201
		M	56.0 ± 14.0	49.7 ± 17.1	45.2 ± 8.3	0.318
	IIB	F	42.8 ± 17.5	45.0 ± 16.4	41.5 ± 14.7	0.401
		M	41.9 ± 13.2	42.7 ± 11.8	45.1 ± 10.5	0.398
ST	I	F	36.9 ± 7.1 ^b^	38.8 ± 7.7 ^ab^	44.1 ± 8.1 ^a^	0.023
		M	58.4 ± 18.1	53.7 ± 20.9	76.8 ± 17.3	0.123
	IIA	F	32.3 ± 9.23	27.8 ± 4.62	44.6 ± 4.1	0.128
		M	39.4 ± 4.3	48.9 ± 13.6	51.7 ± 14.6	0.107
	IIB	F	37.3 ± 9.8	20.1 ± 2.6 ^B^	32.7 ± 3.5	0.248
		M	46.3 ± 11.5	49.8 ± 8.6 ^A^	34.8 ± 5.5	0.128
SM	I	F	46.5 ± 11.9	34.9 ± 10.6	52.0 ± 10.9	0.103
		M	36.0 ± 11.1	53.0 ± 17.5	66.0 ± 24.9	0.121
	IIA	F	33.3 ± 5.1	33.4 ± 5.1	34.5 ± 7.7	0.567
		M	39.1 ± 6.3	30.8 ± 10.3	24.4 ± 4.5	0.341
	IIB	F	37.1 ± 12.2	41.8 ± 9.9	44.5 ± 7.3	0.474
		M	35.4 ± 7.1	36.6 ± 4.5	25.5 ± 4.9	0.343

^a,b^ Means with different letters in the same rows between birth weight groups are significantly different at *p* < 0.05. ^A,B^ Means with different letters in the same column between sexes for each trait are significantly different at *p* < 0.05. Sig. = *p*-value of significant differences, MFT = muscle fibre types, LBW = low birth weight, MBW = medium birth weight, HBW = high birth weight, F = female, M = male, LTL = longissimus thoracis et lumborum, ST = semitendinosus, SM = semimembranosus.

**Table 7 animals-14-00704-t007:** The mean number of Type I, IIA, and IIB fibres/mm^2^ muscle area in LTL, ST, and SM muscles.

Muscles	Fibre Types	Sex	LBW (F; *n* = 8 and M; *n* = 7)	MBW (F; *n* = 7 and M; *n* = 7)	HBW (F; *n* = 7 and M; *n* = 7)
LTL	I	F	883 ± 270	989 ± 197	645 ± 119
		M	1086 ± 327	786 ± 393	422 ± 152
		Sig.	0.217	0.312	0.324
	IIA	F	1120 ± 146 ^B^	1023 ± 181 ^B^	991 ± 162 ^B^
		M	1946 ± 334 ^A^	1706 ± 375 ^A^	2701 ± 660 ^A^
		Sig.	0.042	0.049	0.037
	IIB	F	993 ± 247 ^B^	1645 ± 174	1292 ± 204
		M	2505 ± 643 ^A^	1747 ± 320	1729 ± 239
		Sig.	0.048	0.328	0.342
	Total	F	2996 ± 263 ^B^	3656 ± 246	2928 ± 283
		M	5836 ± 893 ^A^	4137 ± 695	4994 ± 1006
		Sig.	0.042	0.342	0.273
ST	I	F	564 ± 138	613 ± 107	286 ± 41 ^B^
		M	591 ± 129	698 ± 103	918 ± 107 ^A^
		Sig.	0.712	0.697	0.024
	IIA	F	1369 ± 300	1605 ± 750	1939 ± 330
		M	2015 ± 303	973 ± 254	1382 ± 331
		Sig.	0.345	0.432	0.394
	IIB	F	1366 ± 280	2447 ± 430	1370 ± 260
		M	2004 ± 450	2382 ± 446	3315 ± 575
		Sig.	0.398	0.3752	0.207
	Total	F	3299 ± 72	5479 ± 96	3595 ± 76
		M	4610 ± 556	4053 ± 390	5614 ± 804
		Sig.	0.341	0.297	0.245
SM	I	F	278 ± 71	438 ± 95	250 ± 57
		M	292 ± 104	244 ± 77	334 ± 102
		Sig.	0.547	0.105	0.342
	IIA	F	1418 ± 497	1329 ± 310	1337 ± 497
		M	2001 ± 425	2239 ± 722	1555 ± 411
		Sig.	0.341	0.321	0.547
	IIB	F	1677 ± 279	1804 ± 339	1985 ± 471
		M	2213 ± 437	3025 ± 415	2127 ± 550
		Sig.	0.217	0.097	0.476
	Total	F	3453 ± 722	3571 ± 596	3572 ± 512
		M	4505 ± 458	5507 ± 985	4016 ± 435
			0.264	0.452	0.214

^A,B^ Means with different letters in the same column between sexes for each trait are significantly different at *p* < 0.05. Sig. = *p*-value of significant differences, LBW = low birth weight, MBW = medium birth weight, HBW = high birth weight, F = female, M = male, LTL = longissimus thoracis et lumborum, ST = semitendinosus, SM = semimembranosus.

## Data Availability

The data presented in this study are available on request from the corresponding author. The data are not publicly available due to privacy and ethical concerns.

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
