# Peer review of "The Effect of Birth Weight on Fattening Performance, Meat Quality, and Muscle Fibre Characteristics in Lambs of the Karayaka Native Breed"

_animals, 2024, doi:10.3390/ani14050704_

Round 1

Reviewer 1 Report

Comments and Suggestions for Authors

This manuscript studied the effect of birth weight on fattening performance, meat quality, and muscle fiber characteristics in lambs of the Karayaka native breed.

There are comments below:

1.       The introduction should describe and summarize the research progress on the impact of birth weight on growth, meat quality and muscle fiber quantity in sheep, goats, and cattle.

2.       Table 2, Crude protein (%), etc. please give the detailed descriptions of measurement.

3.       Table 2, Crude oil (%)? lease consider if “Crude oil” is a suitable term.

4.       Table 2, Metabolizable energy (kcal/kg), Is this the measured value or the calculated value?

5.       Suggest the authors give more descriptions about the items measurement. For example, the Meat quality analyses, instruments?

6.       Line 246, “blow the previously stated range” Such description is meaningless. What is the previously stated range?

7.       Tables is suggested to give the P values.

8.       The items in the tables are suggested to give the units (some items have units, but some items have no units).

9.       The tables are suggested to give the number of male and female lambs.For example, in the table annotations.

10.   Line 262-273, There is no complete description of the results in Table 4; Line 265-268, How can it be seen that there are no differences? From Table 5, it can be seen that there are differences? Line 268-269, It is not suitable for full names and abbreviations to appear at this time.

11.   Line 297- 299, “Similarly, female 297 lambs with high birth weight had the highest Type I muscle fiber area (P<0.05) in ST mus-298 cle.” How to see 44.1±8.1b38.8±7.7ab36.9±7.1a??

12.   The authors should carefully check all tables and the corresponding result descriptions.

13.   Discussion sector, Please conduct targeted comparisons and discussions based on the relevant results of this study, especially with the studies related to birth weight. For example, line 372-378, “Şen et al. [20] determined the slaughter weight to be 35.6 kg in a study conducted on Karayaka male lambs. The slaughter weight in MBW and HBW groups was found to be higher than in the study conducted by Şen et al. [20]. Higher weights of both warm and cold carcasses were observed in the HBW group in this investigation. The male Karayaka lambs in the HBW group had observations for both warm and cold carcass weights that were greater than those reported by Şen et al. [20] for the same breed.” There are many factors that affect terminal weight, including feed, feeding time, and so on. The authors makes a simple comparison between the results of this article and those of Sen‘s literature. This kind of discussion is puzzling and meaningless. There are still many discussions like this, and the authors need to make significant revisions.

14.   The conclusion section is too lengthy, please refine and shorten it.

Comments on the Quality of English Language

Suggeste native English speaker with professional background to help modify the language to make smoothly.

Author Response

Dear Editor,

Please find attached the revised version of our manuscript (animals-2789424) entitled " The Effect of Birth Weight on Fattening Performance, Meat Quality, and Muscle Fiber Characteristics in Lambs of the Karayaka Native Breed" We would like to take this opportunity to thank the reviewers for their valuable criticisms and contributions, which we believe that they contributed enormously to the scientific quality of our manuscript. We have taken account of all the points that have been raised by the reviewers and have altered our manuscript according to the reviewers’ suggestions. The responses to the comments of the reviewer 1 are as follows.

Comments and Suggestions for Authors

  1. The introduction should describe and summarize the research progress on the impact of birth weight on growth, meat quality and muscle fiber quantity in sheep, goats, and cattle.

Done as requested; please see Lines 77-89 in the revised MS.

  1. Table 2, Crude protein (%), etc. please give the detailed descriptions of measurement.

Done as requested; please see Lines 165-166 in the revised MS.

  1. Table 2, Crude oil (%)? lease consider if “Crude oil” is a suitable term.

Done as requested; please see Lines 165-166 in the revised MS.

  1. Table 2, Metabolizable energy (kcal/kg), Is this the measured value or the calculated value?

This is a calculated value; the relevant explanation has been added to Table 2; please see Lines 165-166 in the revised MS.

  1. Suggest the authors give more descriptions about the items measurement. For example, the Meat quality analyses, instruments?

Done as requested; please see Lines 189-201 in the revised MS.

  1. Line 246, “blow the previously stated range” Such description is meaningless. What is the previously stated range?

The sentences were moved to the discussion section, and this paragraph was revised to avoid confusion; please see Lines 288-302 and Lines 425-432 in the revised MS.

  1. Tables is suggested to give the P values.

Done as requested, P values have been added to all tables in the revised MS.

  1. The items in the tables are suggested to give the units (some items have units, but some items have no units).

Done as requested, missing units in Table 3 have been added in the revised MS.

  1. The tables are suggested to give the number of male and female lambs. For example, in the table annotations.

Done as requested, an explanation of the number of animals has been added in each table annotations; please see Lines 317-319, 351-353, 361-363, 389-391, 400-402, 407-408 in the revised MS.

  1. Line 262-273, There is no complete description of the results in Table 4; Line 265-268, How can it be seen that there are no differences? From Table 5, it can be seen that there are differences?

Done as requested, P values of the results have been added to avoid confusion in Table 4; please see Lines 323-345 in the revised MS.

  1. Line 268-269, It is not suitable for full names and abbreviations to appear at this time.

Done as requested, the full names of colour traits were deleted; please see Lines 339-340 in the revised MS.

  1. Line 297- 299, “Similarly, female lambs with high birth weight had the highest Type I muscle fiber area (P<0.05) in ST muscle.” How to see 44.1±8.1b38.8±7.7ab36.9±7.1a??

Done as requested; sorry for the typing mistake; table 6 was reorganized. P values of the results have been added to avoid confusion. Please see Lines 385-392 in the revised MS.

  1. The authors should carefully check all tables and the corresponding result descriptions.

Done as requested, all tables and the corresponding result descriptions were checked in the results section of the revised MS.

  1. Discussion sector, Please conduct targeted comparisons and discussions based on the relevant results of this study, ………………………………discussions like this, and the authors need to make significant revisions.

Done as requested, the discussion section was reorganized in the revised MS.

  1. The conclusion section is too lengthy, please refine and shorten it.

Done as requested, the conclusion section was refined and shortened; please see Lines 530-547 in the revised MS.

Reviewer 2 Report

Comments and Suggestions for Authors

1. Table 2 what is crude oil? Ether extract? In addition, in the realm of ruminant nutrition, crude fiber is considered outdated, and it is recommended to utilize NDF as a more contemporary alternative. You should specify the measurement methods for these indicators, with a particular emphasis on metabolic energy.

2. Line 231 What is the meaning of the abbreviation SEM?

3. Line 238-239 “However, the initial weight and slaughter weight of male lambs significantly differed from that of female lambs. (P<0.05)”. Has the author compared male lambs with female lambs? Where are the statistical findings? Are different lowercase letters used?

4. Line 240-241 “Furthermore, no variation was observed in the total weight acquired throughout the fattening period (P>0.05) between the groups”. I'm uncertain about the message you're attempting to convey. According to the findings presented in Table 3, there is a notable distinction between the LBW and HBW groups among female lambs. Simultaneously, a significant difference exists between the LBW and MBW groups among male lambs.

5. Line 247-249 The analysis and interpretation provided here are quite perplexing. It's unclear what precisely the author is comparing—whether it's between sexes or between different birth weights. The numerous comparisons in Table 3 only appear to show variations within the same row rather than across columns (i.e., different genders). Additionally, the author failed to elucidate the rationale behind comparing sexes in the statistical analysis outlined in section 2.6.

6. Line 260 only male?

7. Line 261-262 between for 3 groups?

8. Table 5 Why is a variety of data labeled with distinct lowercase letters, despite the absence of any statistical significance?

9. Table 6. The lowercase letter 'a' ought to be assigned to the number that holds the highest value.

10. Line 271-273 Where are the statistical results?

11. Line 297-298 “Similarly, female lambs with high birth weight had the highest Type I muscle fiber area (P<0.05) in ST muscle”. The findings presented in Table 6 contradict your analysis entirely; specifically, female lambs with low birth weight exhibited the highest Type I muscle fiber area.

12. The overall data analysis in the entire paper is perplexing, and there are scant attention-grabbing findings.

Author Response

Dear Editor,

Please find attached the revised version of our manuscript (animals-2789424) entitled " The Effect of Birth Weight on Fattening Performance, Meat Quality, and Muscle Fiber Characteristics in Lambs of the Karayaka Native Breed" We would like to take this opportunity to thank the reviewers for their valuable criticisms and contributions, which we believe that they contributed enormously to the scientific quality of our manuscript. We have taken account of all the points that have been raised by the reviewers and have altered our manuscript according to the reviewers’ suggestions. The responses to the comments of the reviewer 2 are as follows.

Reviewer 2

Comments and Suggestions for Authors

  1. Table 2 what is crude oil? Ether extract? In addition, in the realm of ruminant nutrition, crude fiber is considered outdated, and it is recommended to utilize NDF as a more contemporary alternative. You should specify the measurement methods for these indicators, with a particular emphasis on metabolic energy.

Dear Reviewer, we agree with your criticism. However, the chemical contents of the experimental feeds used in the study were analyzed by the manufacturer. This explanation is stated in the Growth and fattening of lambs subsection of Materials and Methods; please see Lines 161-162 in the revised MS.

  1. Line 231 What is the meaning of the abbreviation SEM?

Done as requested, an explanation of SEM has been added; please see Line 273 in the revised MS.

  1. Line 238-239 “However, the initial weight and slaughter weight of male lambs significantly differed from that of female lambs. (P<0.05)”. Has the author compared male lambs with female lambs? Where are the statistical findings? Are different lowercase letters used?

Done as requested, a sentence was added to the Statistical Analyses subsection of Materials and Methods that compared male lambs with female lambs. Also, differences between sexes were indicated in capital letters in Tables 3, 4, 6, and 7 and P values were added. Please see Lines 270-271 in the revised MS.

  1. Line 240-241 “Furthermore, no variation was observed in the total weight acquired throughout the fattening period (P>0.05) between the groups”. I'm uncertain about the message you're attempting to convey. According to the findings presented in Table 3, there is a notable distinction between the LBW and HBW groups among female lambs. Simultaneously, a significant difference exists between the LBW and MBW groups among male lambs.

This sentence was deleted and replaced with new information has been added to avoid confusion; please see Lines 278-284 in the revised MS.

  1. Line 247-249 The analysis and interpretation provided here are quite perplexing. It's unclear what precisely the author is comparing—whether it's between sexes or between different birth weights. The numerous comparisons in Table 3 only appear to show variations within the same row rather than across columns (i.e., different genders). Additionally, the author failed to elucidate the rationale behind comparing sexes in the statistical analysis outlined in section 2.6.

Table 3 has been rearranged more understandably, and a sentence comparing male lambs with female lambs has been added to the Statistical Analyses subsection of Materials and Methods. Please see Lines 270-271 in the revised MS.

  1. Line 260 only “male”?

Sorry for the typing mistake. The “female” term has been added; please see Line 323 in the revised version.

  1. Line 261-262 “between” for 3 groups?

Done as requested, this sentence was rewritten; please see Line 325 in the revised MS.

  1. Table 5 Why is a variety of data labeled with distinct lowercase letters, despite the absence of any statistical significance?

Done as requested, Table 5 has been rearranged more understandably, differences between storage times were indicated in lower letters in Table 5, and P values were added. Please see Lines 358-364 in the revised MS.

  1. Table 6. The lowercase letter 'a' ought to be assigned to the number that holds the highest value.

Done as requested; please see Table 6 Lines 385-393 in the revised MS.

  1. Line 271-273 Where are the statistical results?

Done as requested, a sentence was added to the Statistical Analyses subsection of Materials and Methods that compared storage times. Also, differences between storage times were indicated in lower letters in Table 5, and P values were added. Please see Lines 270-271 and 358-361 in the revised MS.

  1. Line 297-298 “Similarly, female lambs with high birth weight had the highest Type I muscle fiber area (P<0.05) in ST muscle”. The findings presented in Table 6 contradict your analysis entirely; specifically, female lambs with low birth weight exhibited the highest Type I muscle fiber area.

Sorry for the typing mistake, Table 6 was reorganized. Please see Lines 385-392 in the revised MS.

  1. The overall data analysis in the entire paper is perplexing, and there are scant attention-grabbing findings.

Done as requested; tables and the results were re-written more clearly and understandable in the revised MS.

Reviewer 3 Report

Comments and Suggestions for Authors

Dear sir

may be modified as suggested

Comments on the Quality of English Language

Dear sir

may be modified as suggested

Author Response

Dear Editor,

Please find attached the revised version of our manuscript (animals-2789424) entitled " The Effect of Birth Weight on Fattening Performance, Meat Quality, and Muscle Fiber Characteristics in Lambs of the Karayaka Native Breed" We would like to take this opportunity to thank the reviewers for their valuable criticisms and contributions, which we believe that they contributed enormously to the scientific quality of our manuscript. We have taken account of all the points that have been raised by the reviewers and have altered our manuscript according to the reviewers’ suggestions. The responses to the comments of the reviewer 3 are as follows.

Review 3

Comments and Suggestions for Authors in the Attached File

  1. Line 29 - “Ad libitum access to feed and water was provided…………. study's conclusion.” modified as “Throughout the fattening phase, the lambs were given ad libitum access………….of the study.”

Done as requested; please see Lines 29-30 in the revised MS.

  1. Values related to the study may also be provided.

Done as requested; please see Lines 40, 46, and 47 in the revised MS.

  1. Lines 57-58 - “Türkiye has 58 approximately …… sheep industry's existence is 59 developing the native breeds.” modified as “There are over 42 million sheep in ………. sheep business depends heavily on the advancement of the local breeds.”

Done as requested; please see Lines 61-64 in the revised MS.

  1. Lines 109-110 - The study was done during…..Gaziosmanpasa University, Tokat, Türkiye.” modified as “The experiment was carried out at the …….. the breeding season (September to March).”

Done as requested; please see Lines 126-129 in the revised MS. 

  1. Line 126 - “Two weeks after birth, ewes were…… the daytime.” modified as “Two weeks after birth, ewes were allowed to pasture for grazing during the daytime.”

Done as requested; please see Lines 145-146 in the revised MS.

  1. “ad libitum” Italics in all places.

Done as requested; "ad libitum" has been redacted throughout the text in italics; please see Lines 30, 153, and 159 in the revised MS.

  1. Lines 132-133 - “All lambs were weaned………………. following day at weaning.” modified as “At three months of age, all lambs were…………………………………next day at weaning.”

Done as requested; please see Lines 154-158 in the revised MS.

  1. This is written like discussion in multiple places and the complete results portion may be condensed and only the pattern of results and level of significance may be presented and avoid repeating the values given in the table.

Done as requested; only the results and significance level pattern were presented in the results section; please see Lines 279-285, 295-302, 327-334, and 380-383 in the revised MS.

  1. Number of observations may be given for all tables

Done as requested, an explanation of the number of animals has been added in each table annotations; please see Lines 317-319, 351-353, 361-363, 389-391, 400-402, 407-408 in the revised MS.

  1. Values related to correlation between birth weight and other traits in your study may be given

We thank the reviewer very much for that excellent point. We guarantee that this valuable advice will be considered in our future work. 

  1. Lines 401-409 -Rewrite and avoid using L*, a*, and b* in running text

Done as requested; please see Lines 467-476 in the revised MS. 

  1. “positive correlation between birth weight and warm and cold carcass weights.” No value given and may be provided

The sentence was revised to avoid confusion; please see Lines 413-415 in the revised MS. 

  1. May be shortened and more precise with related to the results.

Done as requested, the conclusion section was refined and shortened; please see Lines 530-547 in the revised MS.

Round 2

Reviewer 1 Report

Comments and Suggestions for Authors

This manuscript studied the effect of birth weight on fattening performance, meat quality, and muscle fiber characteristics in lambs of the Karayaka native breed.

 There are comments below:

1.       In Table 2, please note the correctness of using crude oil to express crudefat.

2.       Line 162, Analyzed should not be used here, it can be said that the company provides. Due to metabolic energy, the company may only have calculated metabolic energy and needs to confirm whether this is a calculated value or a measured value. If it is a calculated value, please indicate below the table that metabolic energy is the calculated value.

3.       Line 201, WHC, Abbreviations suddenly appear?

4.       Line 271, Firstly, pH values were not compared between 1 and 24 hours, only gender comparison was conducted. Secondly, it is not recommended to compare color parameters between groups at 1h and 24h. It is more appropriate to compare the gender of each group at 1h or 24h, which is more consistent with the statistics of the entire text.

5.       Tables, “a,b Means with different letters in the same rows are signifi-314 cantly different at P<0.05 or P<0.01. “ ab can simultaneously represent 0.05 or 0.01?

6.       Tables, “*See Table 1 in Animals subsection of Materials and Methods for the numbers of 318 examined lambs in birth weight groups.”  This description is meaningless. The table should be self explanatory, please provide information on animals in the header or table.

7.       Line 380-381, This is only the number of type 2 muscle fibers in LD muscles, males are greater than females.

8.       Line 493, this study, does this refer to this present study or Dwyer’s research?

Comments on the Quality of English Language

The language can continue to be further optimized.

Author Response

Dear Reviewer,

Attached is the revised 2 version of our manuscript (animals-2789424) entitled " The Effect of Birth Weight on Fattening Performance, Meat Quality, and Muscle Fiber Characteristics in Lambs of the Karayaka Native Breed." We have considered all the points you have raised and have altered our manuscript according to your suggestions. The responses to the comments of the reviewer are as follows:

Comments and Suggestions for Authors
1. In Table 2, please note the correctness of using crude oil to express crudefat.
Done as requested; “crude oil” changed to “crude fat”; please see Line 163 in the revised 2 MS.

2. Line 162, Analyzed should not be used here, it can be said that the company provides. Due to metabolic energy, the company may only have calculated metabolic energy and needs to confirm whether this is a calculated value or a measured value. If it is a calculated value, please indicate below the table that metabolic energy is the calculated value.
Done as requested; please see Lines 159-160 and 164-169 in the revised 2 MS.

3. Line 201, WHC, Abbreviations suddenly appear?
Done as requested; WHC was deleted and an explanation of the WHC abbreviation has been added to the text; please see Lines 205-206 in the revised 2 MS.

4. Line 271, Firstly, pH values were not compared between 1 and 24 hours, only gender comparison was conducted. Secondly, it is not recommended to compare color parameters between groups at 1h and 24h. It is more appropriate to compare the gender of each group at 1h or 24h, which is more consistent with the statistics of the entire text.
Done as requested; Tables 4 and 5 were reorganized according to the study's statistics. Please see Lines 340-341 and 353-354 in the revised 2 MS.

5. Tables, “a,b Means with different letters in the same rows are signifi-314 cantly different at P<0.05 or P<0.01. “ ab can simultaneously represent 0.05 or 0.01?
Done as requested; “P<0.01” was deleted to avoid confusion in the Statistical analyses section and Table 3; please see Lines 276 and 310 in the revised 2 MS.

6. Tables, “*See Table 1 in Animals subsection of Materials and Methods for the numbers of 318 examined lambs in birth weight groups.”  This description is meaningless. The table should be self-explanatory; please provide information on animals in the header or table.
Done as requested, the description of the number of animals below each table was deleted, and the animal numbers were added to the Tables. Please see Lines 306, 340, 353, 383, 395, and 405 in the revised 2 MS.

7. Line 380-381, This is only the number of type 2 muscle fibers in LD muscles, males are greater than females.
Done as requested; the sentences were rearranged. Please see Lines 378-380 in the revised 2 MS.

8. Line 493, this study, does this refer to this present study or Dwyer’s research?
Done as requested; the sentence was rearranged. Please see Lines 481 and 487 in the revised 2 MS.

9. The language can continue to be further optimized.
Done as requested; we have improved the language of the revised MS as much as possible and are ready to improve more if required.

Reviewer 3 Report

Comments and Suggestions for Authors

The article has been improved and the significance value given in table in rows and columns are little confusing may be suitably explained or just indicating with letters alone is enough. If possible correlation between the weight with different parameters may be included 

Author Response

Dear Reviewer,

Attached is the revised 2 version of our manuscript (animals-2789424) entitled " The Effect of Birth Weight on Fattening Performance, Meat Quality, and Muscle Fiber Characteristics in Lambs of the Karayaka Native Breed." We have considered all the points you have raised and have altered our manuscript according to your suggestions. The responses to the comments of the reviewer are as follows:

Comments and Suggestions for Authors
1. The article has been improved and the significance value given in table in rows and columns are little confusing may be suitably explained or just indicating with letters alone is enough. If possible correlation between the weight with different parameters may be included
Table 4 and Table 5 have been rearranged to make the differences between birth weights and sexes more understandable. Relationships between variable characteristics related to birth weight were determined by Pearson correlation analysis and added to the statistical analysis section. As a result of the analysis, no relationship was found between variable characteristics related to birth weight, which is stated in the results and discussion sections. Please see Lines 274-275, 289-292 and 379-381 in the revised 2 MS. Also, the file of correlation analysis results is attached.
